# Selection of Superior *Senna macranthera* Seeds, Carbon Stock, and Seedling Survival, and Costs for Habitat Restoration

**Joyce de Oliveira Araújo** [1,*], **Daniel Teixeira Pinheiro** [1], **Geovana Brito Queiroz** [1], **Júlia Martins Soares** [1], **Aaron Kinyu Hoshide** [2,3], **Vicente Toledo Machado de Morais Junior** [4], **Samuel José Silva Soares da Rocha** [5] and **Denise Cunha Fernandes dos Santos Dias** [1]

1. Agronomy Department, Laboratory of Seed Analysis, Universidade Federal de Viçosa, Viçosa 36570-900, MG, Brazil
2. College of Natural Sciences, Forestry and Agriculture, The University of Maine, Orono, ME 04469, USA
3. AgriSciences, Instituto de Ciências Agrárias e Ambientais, Universidade Federal de Mato Grosso, Avenida Alexandre Ferronato, 1200, Sinop 78555-267, MT, Brazil
4. Brandt Meio Ambiente LTDA, Alameda do Ingá, 89, Vale do Sereno, Nova Lima 34006-042, MG, Brazil
5. Departamento de Ciências Florestais, Universidade Federal de Lavras, Lavras 37200-900, MG, Brazil
* Correspondence: joyce.araujo@ufv.br; Tel.: +55-(31)-3899-2619

**Abstract:** Conservation and recovery of degraded areas generate great demand for seeds of native tree species. The development and/or improvement of efficient techniques for the evaluation of forest-seed quality is important for the production and establishment of high-quality seedlings for restoration. In this study, the tissue density of radiographic images of *Senna macranthera* seeds was related to their physiological quality. Moreover, biomass, carbon stock, seedling survival, and X-ray technique costs were estimated for *S. macranthera*. Collected seeds were analyzed using digital radiography to measure relative and integrated density. The physical integrity of seed tissues was visually evaluated. Seeds were then germination tested to assess seedling development-related traits. Semiautomated radiography allowed for visualizing internal seed structures and observing their density and physical-integrity differences as well as physiological quality. Moreover, seed lots with lower relative and integrated densities had more physical damage and/or malformation, thus producing less vigorous seedlings. The average carbon stock was 21.42 kg per tree. The seed selection cost was USD 0.0132/seed at an 81% germination rate. The annual cost of planting *S. macranthera* seedlings was USD 7500 per hectare during the establishment year and averaged USD 1562 per year for replanting lost transplants over the eight years after initial planting. Applying these techniques may enhance the seedling production of this species, contributing to reforestation programs in Brazil.

**Keywords:** biomass; forest species; integrity; semiautomated analysis; *Senna macranthera*; X-rays

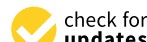



## 1. Introduction

*Senna macranthera* (DC. ex Collad.) H.S. Irwin & Barneby is a tree species native to Brazil, belonging to the family Fabaceae-Caesalpinioideae. In Brazil, its occurrence is common from Ceará to São Paulo states, occupying Caatinga, Atlantic Forest, and Cerrado biomes [1]. *S. macranthera* is a rustic and fast-growth species, suitable for mixed plantations used for the restoration of degraded rural areas [2], in addition to being used for urban afforestation [3]. Programs for the conservation and recovery of degraded areas generate great demand for seeds of native tree species [4]. In this context, studies on the development and/or improvement of efficient techniques for the evaluation of forest-seed quality are important for the production and establishment of high-quality seedlings.

Forest-seed quality analysis using X-rays has generated consistent, rapid, and nondestructive results. This process involves X-ray absorption by seed tissues, which are reflected in grayscale radiographic images [5]. Factors such as tissue composition, thickness, and density, in addition to ionizing radiation wavelength, interfere with the level of X-ray

absorption by seeds, generating images with greater or lesser degrees of radiopacity (light) and radioluminescence (dark) [6]. X-ray testing is recommended by the International Seed Testing Association for the detection of full, empty, mechanically damaged, or insect-attacked seeds [7]. As it is a nondestructive method, the same seeds can be further used in physiological tests for internal morphology, relating results to seedling development [8]. Furthermore, this test is important in selecting undamaged seeds to form high-quality seed lots, which in turn are crucial in ex situ germplasm conservation [9].

New approaches have been employed in X-ray testing for seed-quality assessment to automate seed radiography analysis. Semiautomated analysis of digital radiographs can relate seed internal morphology and tissue integrity which can provide potential parameters associated with physiological seed quality [10,11]. Such parameters can be measured by specific image-analysis software, making the interpretation of results faster and less subjective, and defining standardized evaluation criteria, which can be more accurate than evaluations based on human vision alone.

ImageJ$^®$ is an open-source software used for digital-image processing [12]. It has shown potential for analysis of seed radiographs [10,13–15]. This software can measure morphometric traits (area, perimeter, circularity, and others) and radiographic-image pixel density, using variables such as relative and integrated densities. Recent studies have shown that such parameters can be used to make inferences about seed internal-tissue density related to seed physiological quality for different species, such as *Leucaena leucocephala* [11] and *Piptadenia gonoacantha* [16].

Using modern software and technologies to select superior-quality seeds can improve both access to and the quality of planting material. This can contribute to restoring Brazilian biomes. During the 2015 Paris Agreement on climate change, Brazil committed to restoring 12 million hectares of degraded areas with forests by 2030 [17]. According to Trujillo-Miranda et al. (2018), there is great urgency to restore these areas, however, the costs involved in forest restoration are high [18]. To increase the probability of success during forest-restoration and carbon-offset projects, it is essential to understand both the performance of native species [19] as well as the quality of seeds that are used [20]. However, there is limited information on *Senna macranthera* seed technology, carbon stock, biomass estimates, and costs of using this tree species for reforestation.

There is great potential for using digital radiographic images to assess seed quality in a semiautomated way. Therefore, the goal of this study was to use such radiographic images of *Senna macranthera* seeds to determine the relationship between tissue-density measures and physiological seed quality. This process has the potential for future commercialization for improving the seed quality of *Senna macranthera* for use in reforestation projects. Therefore, we estimated *S. macranthera* biomass, carbon stock, seedling survival, and seedling cost for the recovery of degraded areas and for reforestation. We then estimated the cost of selecting higher-quality *S. macranthera* seeds using this X-ray technique. We hypothesize that *S. macranthera* seed selection can be cost-effective, which can encourage greater use of this species in reforestation and carbon sequestration efforts in Brazil.

## 2. Materials and Methods

### 2.1. Study Area and Plant Seeds

The biological nomenclature used for *Senna macranthera* was from the International Plant Names Index [21] with proper formatting from Peruzzi (2020) [22]. The study was carried out at the Laboratory of Seed Research, Department of Agricultural Science, Federal University of Viçosa, Minas Gerais State, Brazil. *S. macranthera* seeds were collected, in September 2021, from ripe fruit from the midregion of the canopy of ten mother trees (8.5 years old) constituting ten seed lots. All trees were located in the city of Viçosa (20°43′40.2″ S, 40°50′41.9″ W). Initially, the water content of each seed lot was determined by the oven method at 105 °C for 24 h [6]. To do so, four 25-seed replications were used for each lot, and results were expressed as a percentage (wet basis).

### 2.2. Acquisition and Processing of Radiographic Images

Radiographic images were obtained using four 25-seed replications per treatment. Treatments consisted of 10 lots of seeds. The seeds were placed in an orderly manner on adhesive paper to allow for the identification of individual seeds. Digital radiographic images were generated using a Faxitron MX-20 digital radiography system (Faxitron X-ray Corp. Wheeling, IL, USA), which was configured at 10 s radiation exposure, 32-kilovolt (kV) voltage, 27.8-cm focal length, and image contrast (unitless) calibrated at 3792 (width) × 5374 (center). After the acquisition, images were saved in TIFF format, following semiautomatic processing and evaluation using ImageJ$^®$ software. This is how the variables' relative density (Dr) and integrated density (Di) were obtained. They were respectively defined as the sum of gray values of all pixels within the selected area divided by the number of pixels therein (expressed as gray/pixel), and the sum of pixel values within an image or selection area (expressed as gray mm$^2$/pixel).

To better interpret and understand the relationship between tissue density from radiographic images and seed physiological quality, relative and integrated densities were classified as a function of the frequency of values observed for seeds from each lot. Thus, for relative density, three classes were adopted, namely: CI—densities above 143 gray/pixel; CII—from 118.1 to 143 gray/pixel; CIII—below or equal to 118 gray/pixel. Yet, for the integrated density, classes were: CI—above 4064 gray mm$^2$/pixel; CII—from 2765.1 to 4064 gray mm$^2$/pixel; CIII—below or equal to 2765 gray mm$^2$/pixel.

### 2.3. Visual Analysis of Radiographs

From visual radiographic analysis, the rate of seeds with severe and mild alterations in tissue physical integrity was determined. Severe alterations consisted of seeds with a compromised embryonic axis or with more than 50% of their area altered by any type of physical damage (insect and/or mechanical) or malformation. On the other hand, mild alterations were defined as seeds with cotyledons with less than 50% of their area damaged and no embryonic axis damage.

### 2.4. Seed Physiological Quality

After obtaining radiographic images, the seeds were manually scarified (with sandpaper number 100) on the opposite side of the embryonic axis to allow water entry. Then, they were subjected to a germination test. To this end, seeds were distributed on moistened paper towels, following the order of image acquisition to enable an association between germination and X-ray findings. Paper sheets were moistened with 2.5 times their dry weight in water. After sowing, rolls were made and then kept in a germinator at 25 °C. Germinated seeds were counted daily until 7 days after the test installation [3]. Results were expressed as a percentage of normal seedlings determined at the final count on the 7th day. Moreover, the root protrusion index was calculated using the GermCalc function of the SeedCalc package of the R software, version 4.1.2 [23], which considered seedlings with primary roots larger than 2 mm.

After the germination test, seedlings were photographed next to a size reference (cm-graduated ruler) for further semiautomatic analysis of growth. Furthermore, after drying in a forced air circulation oven at 80 °C for 24 h, seedling dry mass (expressed as mg/seedling) was determined according to methods outlined in Krzyzanowski et al. (2020) [24]. Seedling-length images were taken in a closed box made of medium-density fiberboard (MDF) with a copy stand attached and a photographic base (blue ethylene–vinyl acetate sheet), which was under 340-lux light intensity. The images were acquired by a Canon EOS Rebel T7+ digital camera, using a 1/80 s shutter speed and f/4.0 aperture. The camera was kept at a height of 50 cm and an angle of 90° to the photographic base. The images were processed semiautomatically by the ImageJ$^®$ software. Using results expressed in millimeters per seedling, uniformity [25] and corrected vigor [5] indexes were generated, with the help of the PlantCalc function of the SeedCalc package and equations proposed by Silva et al. (2019) [23].

### 2.5. Experimental Design and Statistical Analysis

The experiment was carried out in a completely randomized design, with four replications. Data were subjected to analysis of variance (ANOVA). After confirming normal error distribution by the Shapiro–Wilk test and variance homogeneity by Bartlett's test, the means were compared by the Scott–Knott test at 5% probability. A multivariate principal component analysis (PCA) was also performed between the data obtained from seed radiography and physiological quality evaluations. The R statistical software, version 4.1.2, was used for all analyses.

### 2.6. Estimation of Biomass, Carbon Stocks, and Survival Rate

To evaluate the potential of *Senna macranthera* in reforestation projects, we analyzed data from the species in an experimental planting. The study was carried out at the Federal University of Viçosa, Minas Gerais State, Brazil. Data on plant survival, biomass, and carbon storage were collected through nondestructive collections according to the methods outlined in Morais Junior et al. (2019) [26]. Census-type forest inventories were conducted at 42, 66, 90, and 102 months. The diameters at ground level (DGL) were collected in millimeters (mm) using a precision digital caliper up to a diameter of 20 mm and, above that, a tape measure was used. The total height (H) in centimeters (cm) was obtained using a graduated rod up to 7 m (m) with measurements above this height using the Vertex Laser 5. The average carbon of the stem and branches ($\mu$C) was estimated from the tree volume, biomass, and carbon of the trees. The volume of each section (lower, middle, and upper third of the stem, and branches) was obtained using the Smalian equation:

$$Vj = ((As1 + As2)/2) \times L)) \tag{1}$$

where:

Vj = volume of the jth section in m$^3$;
As1 = initial sectional area (m$^2$);
As2 = final sectional area (m$^2$);
L = longitudinal section length (m).

The sum of the volumes of the sections of the jth individual section constituted the total volume of each plant. The biomass of the branches and stem of each plant was obtained by multiplying the basic density of the wood by the individual total volume. The basic density of each plant was estimated according to a bibliographical review of these species/genera in Brazil [27]. The conversion of carbon into biomass was performed assuming a carbon proportion of 0.48 relative to the woody biomass [28]. A carbon-storage equation of tree trunks and branches [29] generated by Morais Junior et al. (2019) [20] was used:

$$Cij = [0.000353 \times (DGL^{1.202424}) \times (H^{0.781883})] \text{ (adjusted R}^2 = 82.12\%) \tag{2}$$

where: Ci = carbon stock of the ith individual (kg);

DGL = diameter at ground level (mm);
H = height (m).

### 2.7. Estimation of Seedling Survival Rate, Replanting Costs, and Seed Selection

Survival rates of this species were used to evaluate the efficiency of using this species in terms of costs for replanting seedlings. The cost of a *Senna macranthera* seedling was considered to be around USD 3.10 (base year December 2022 where the USA dollar was valued at 5.15 reais (BRL) of Brazilian currency: https://g1.globo.com/economia/noticia/2022/12/23/dolar-opera-em-queda-e-comeca-o-dia-a-r-517.ghtml, accessed on 19 January 2023). Replanting costs were calculated as just the cost of seedlings and not labor. The cost of planting a uniform stand of *Senna macranthera* seedlings was calculated with an assumed seedling planting density of 2500 seedlings per hectare (ha). Transplanting occurred when seedlings were 3 months old. Seedling mortality was measured and any dead plants

were replaced with new 3-month-old seedlings. Replanting was done at 42, 66, 90, and 102 months (3.5, 5.5, 7.5, and 8.5 years) after the original establishment of the planting.

Currently, there is no formal market for *Senna macranthera* seeds. Therefore, we estimated the total cost of processing these seeds to select lots with high physical and physiological quality following our specific laboratory methods. Total costs included both variable costs, which change with the level of seed produced, as well as fixed costs which do not change over the annual time frame of our budget. Variable costs included hiring labor at USD 4.50 per hour to run the X-ray machine and camera for imaging as well as analyzing and separating viable from nonviable seeds. Fixed costs include lab rental at USD 700 per month plus depreciation of the purchase value (USD) of the X-ray (25,000), camera (700), and computer (1000) used for seed imaging and analysis. All equipment was assumed to have a useful life of 15 years and 0% salvage value.

## 3. Results

### 3.1. Radiography Analyses

The seed moisture content was similar among analyzed lots, ranging between 9.5 and 10.3% (Figure 1A). This moisture range allowed clear visualization of seed internal morphology and structures (integument, cotyledons, and embryonic axis; Figure 1B). Within this range, we could identify malformed seeds (Figure 1C), physical damage by predation (Figure 1D), and embryonic axis mechanical damage (Figure 1E).

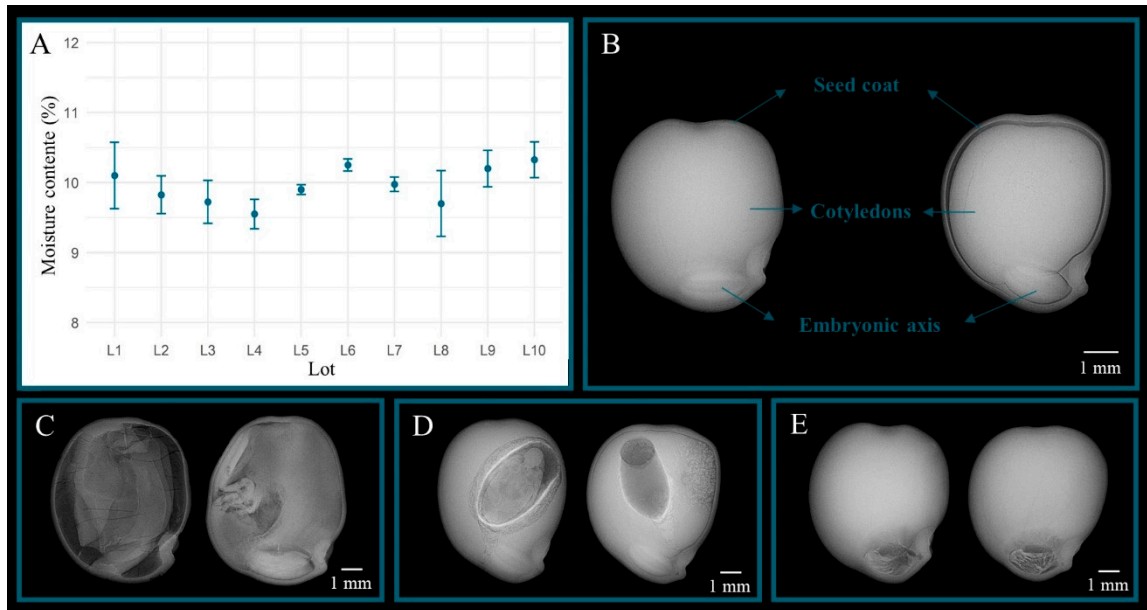

**Figure 1.** (**A**) Seed moisture content in the 10 lots of *Senna macranthera* seeds. Radiographic images of seeds showing (**B**) internal morphology and structures, (**C**) malformation, (**D**) insect predation damage, and (**E**) embryonic axis damage. Bars indicate standard errors.

Figure 2 shows the significant differences among *Senna macranthera* seed lots for density by the ImageJ® software. Both for relative (Figure 2A) and integrated (Figure 2B) densities, lots 1, 2, 5, 7, and 9 had the highest values, while lot 8 had the lowest values. Overall, the seeds from lots 3, 4, 6, and 10 showed intermediate tissue densities.

Seed division into relative density classes showed that lot 2 had a higher percentage of denser seeds (CI; 71%) compared to the others (Figure 3A). On the other hand, lot 8 had the lowest percentage of denser seeds (CI; 32%) and the highest of less-dense seeds (CIII; 27%). Lot 1, despite having a lower percentage of denser seeds (CI) compared to lot 2, had only 1% of low relative-density seeds (CIII). Considering integrated density classes (Figure 3B), all lots had higher proportions of intermediate-class seeds (CII). However, as observed for

relative density (Figure 3A), lot 8 had the highest percentage of less-dense seeds (CIII; 34%) (Figure 3B).

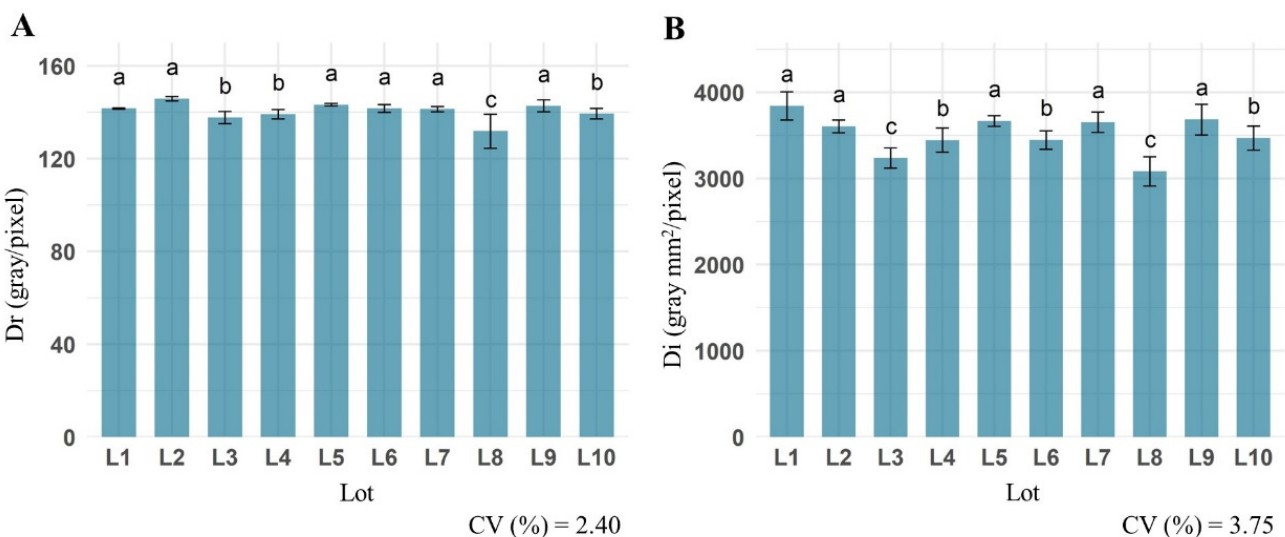

**Figure 2.** (**A**) Relative density and (**B**) integrated density of different lots of *Senna macranthera* seeds obtained using the ImageJ® software. Equal letters do not differ by the Scott-Knott test ($p \leq 0.05$). Bars indicate standard deviation.

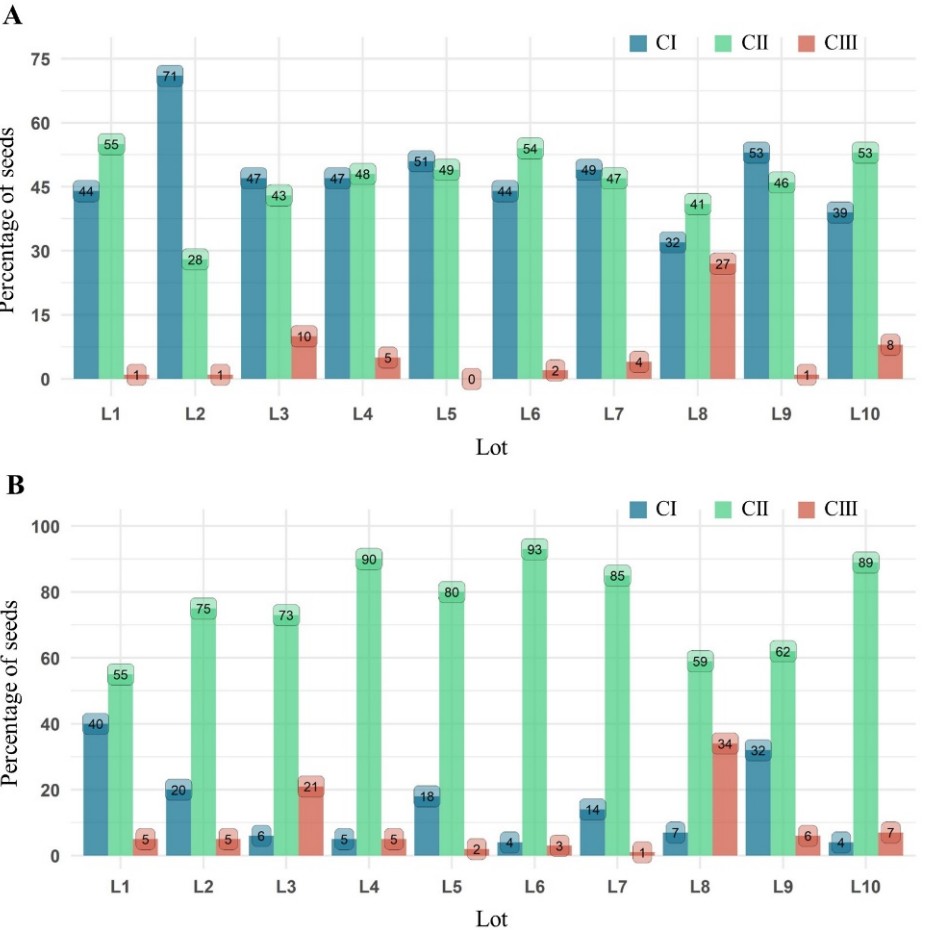

**Figure 3.** Percentage of *Senna macranthera* seeds in different classes of (**A**) relative density and (**B**) integrated density.

Figure 4 highlights the direct relationship between radiographic images and their relative frequency histograms of the density of *Senna macranthera* seeds. In short, CI seeds (higher density) are more filled and have no mechanical damage, with pixels concentrated in higher gray-level regions of the histogram (about 255 pixels). On the other side, CII seeds (intermediate density) show good filling but also physical damage (e.g., insect injuries), decreasing their radiation absorption. Finally, CIII seeds (lower density) have more translucent or less darkened tissues, in addition to more physical damage, with histogram peaks in lower gray-level regions compared to other regions.

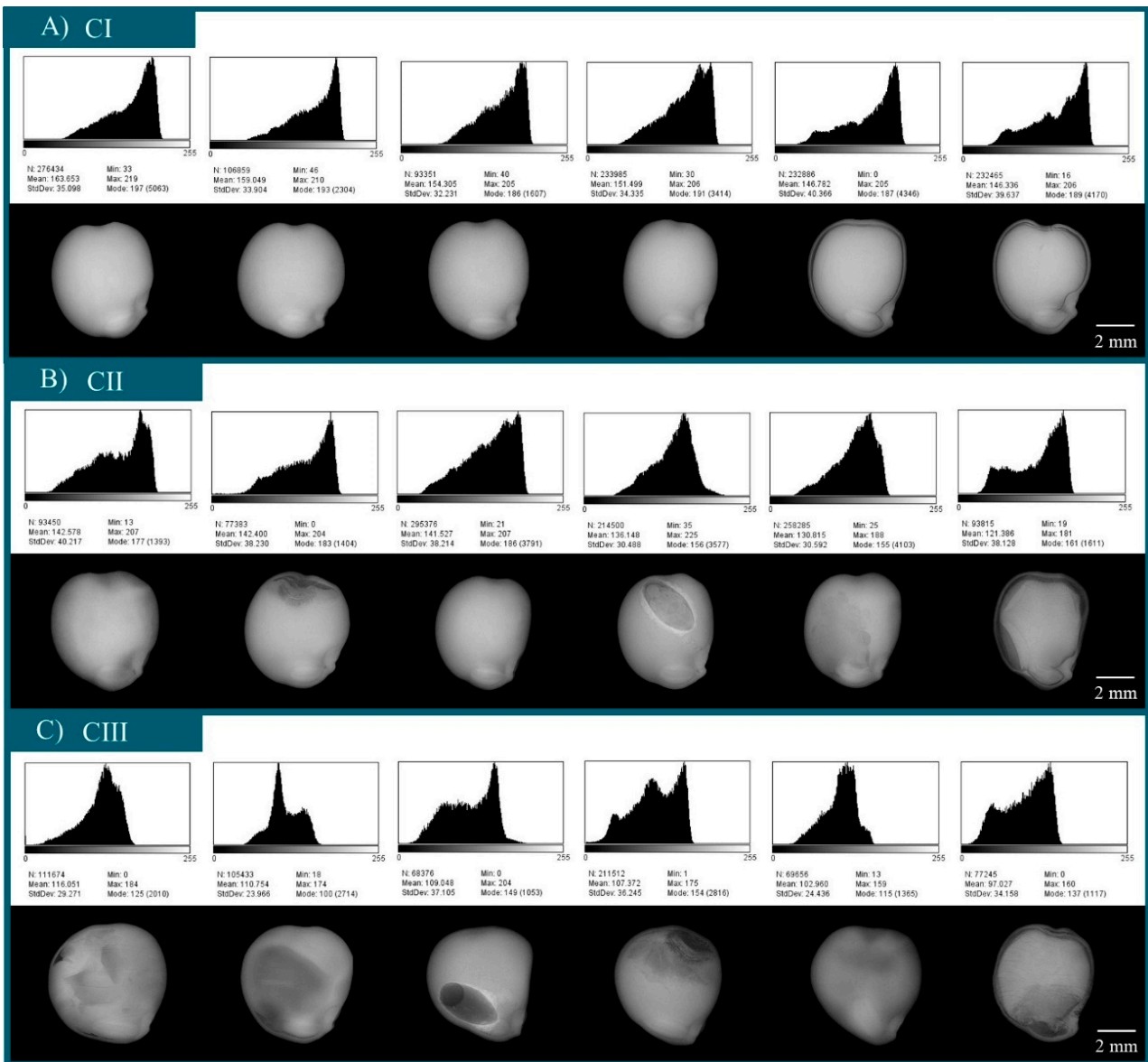

**Figure 4.** Illustration of seeds from the three classes and their relative density histograms. (**A**) CI: relative density above 143 gray/pixel; (**B**) CII: densities from 118.1 to 143 gray/pixel; (**C**) CIII: densities below or equal to 118 gray/pixel.

Visual analysis of radiographs allowed us to identify changes in the seed tissue's physical integrity. Through that, we could observe the presence or absence of damage, as well as its location, intensity, and area. Alterations were more severe in lot 8, followed by lots 3, 4, and 10. Therefore, lot 8 had the highest proportion of seeds damaged along the embryonic axis and seeds with more than 50% of their area damaged. Regarding mild changes and the absence of damage, lots 1, 2, 5, 6, 7, and 9 showed no significant differences.

Therefore, these lots had a lower percentage of cotyledon-damaged seeds (<50% of the seed area) and were superior in terms of seed-tissue physical integrity (Table 1).

Regarding seed physiological quality, lots 1, 2, and 9 showed the highest germination rates (95%, 94%, and 92%, respectively), whereas lots 3, 4, 8, and 10 had the lowest ones (69%, 71%, 46%, and 74%, respectively). In terms of seed vigor, seedling performance, root protrusion speed index (RPSI), seedling length (SL), corrected vigor index (CVI), and seedling dry matter (SDM) was higher in lots 1 and 2 compared to other lots. However, there was no significant difference among lots for uniformity index (UI). Concerning seedling performance, lot 8 showed the lowest values for all variables analyzed. Despite having a high germination rate, lot 9 had lower seedling growth (hence lower seed vigor) compared to lots 1 and 2. Regarding seed germination and vigor results, lots 3, 4, 5, 6, 7, and 10 showed, on average, an intermediate physiological potential among all lots analyzed (Table 2).

**Table 1.** Incidence of alterations in tissue physical integrity of *Senna macranthera* seeds from 10 lots.

| Lot | Changes in Tissue Physical Integrity | | |
| | Severe | Mild | Absent |
| | ---------------------- (%) ---------------------- | | |
| 1 | 2 d | 2 b | 96 a |
| 2 | 3 d | 4 b | 93 a |
| 3 | 22 b | 15 a | 63 c |
| 4 | 15 c | 18 a | 67 c |
| 5 | 1 d | 9 b | 90 a |
| 6 | 7 d | 5 b | 88 a |
| 7 | 6 d | 8 b | 86 a |
| 8 | 34 a | 17 a | 49 d |
| 9 | 5 d | 6 b | 89 a |
| 10 | 13 c | 14 a | 73 b |
| F | 36.73 * | 9.25 * | 37.29 * |
| CV (%) | 32.08 | 38.72 | 6.41 |

Means followed by the same letter in the column do not differ statistically by the Scott–Knott test. ($p \leq 0.05$). * = Significant; F = calculated F-value; CV = coefficient of variation.

**Table 2.** Initial characterization of the physiological quality of *Senna macranthera* seeds from 10 lots.

| Lot | Germination (%) | Root Protrusion Speed Index | Seedling Length (mm/Seedling) | Uniformity Index | Corrected Vigor Index | Seedling Dry Matter (mg/Seedling) |
|---|---|---|---|---|---|---|
| 1 | 95 a | 7.65 a | 140.74 a | 874.60 | 717.21 a | 71.84 a |
| 2 | 94 a | 7.83 a | 109.17 b | 842.19 | 596.67 b | 71.70 a |
| 3 | 69 c | 5.71 c | 88.94 d | 764.43 | 373.04 d | 31.20 b |
| 4 | 71 c | 5.69 c | 81.20 d | 728.19 | 356.91 d | 68.09 a |
| 5 | 88 b | 6.93 b | 93.63 c | 774.26 | 492.78 c | 71.55 a |
| 6 | 87 b | 7.05 b | 92.13 c | 778.99 | 483.83 c | 71.46 a |
| 7 | 87 b | 6.80 b | 87.13 d | 718.61 | 452.88 c | 71.14 a |
| 8 | 46 d | 3.60 d | 71.16 d | 646.52 | 204.60 e | 20.54 c |
| 9 | 92 a | 7.11 b | 96.33 c | 768.55 | 522.29 c | 71.16 a |
| 10 | 74 c | 5.85 c | 85.99 d | 707.08 | 379.68 d | 70.73 a |
| F | 43.37 * | 21.59 * | 18.41 * | 1.94 ns | 38.44 * | 214.19 * |
| CV (%) | 5.81 | 8.39 | 9.31 | 12.42 | 9.97 | 4.24 |

Means followed by the same letter in the column do not differ statistically by the Scott–Knott test ($p \leq 0.05$). * = significant; ns = nonsignificant; F = calculated F-value; CV = coefficient of variation.

Principal component analysis (PCA) demonstrated that components 1 (PC1) and 2 (PC2) explained 84.4% of total data variability (Figure 5A). The correlation circle (Figure 5B) revealed that lots 1, 2, 5, 6, 7, and 9 were concentrated in positive PC1 scores,

near germination (G; orange color), vigor (RPSI, SL, UI, CVI, and SDM; green color), and CI class vectors. These results indicate higher integrated tissue density (CI−Di; red color) and relative tissue density (CI−Dr; blue color). Otherwise, lots 3 and 8 had negative PC1 scores of the ordering diagram, far from physiological quality-related vectors but close to class CIII vectors, thus having a lower tissue density (CIII−Di and CIII−Dr). Finally, lots 4, 5, 6, 7, and 10 dispersed intermediately compared to the other lots (Figure 5A). The vector that composes the absence of alteration in tissue physical integrity was close to the vectors density and physiological quality, with high correlation, especially with G, RPSI, Dr, and Di. On the other hand, the vector for severe alteration was in the opposite quadrant (PC1-/PC2+) and closer to vectors related to the CIII class of the relative and integrated densities (Figure 5B).

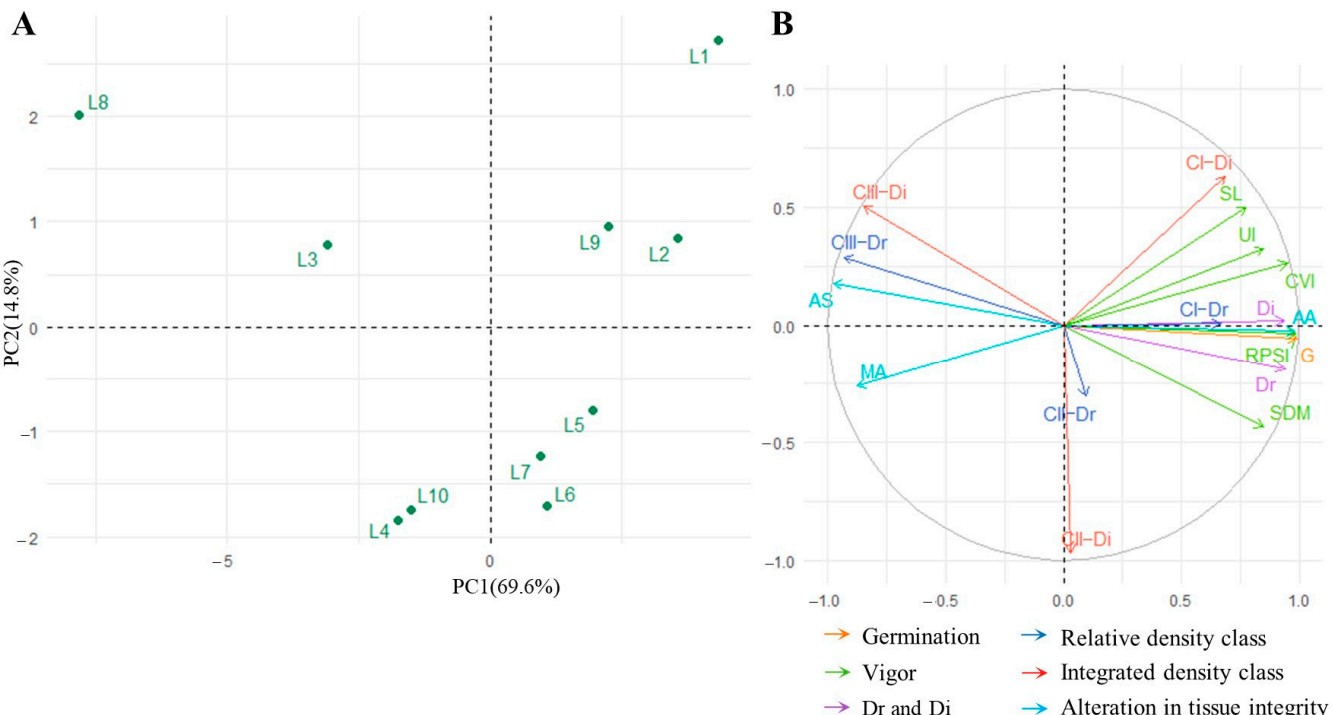

**Figure 5.** (**A**) Ordering diagram and (**B**) correlation circle obtained by principal component analysis (PCA). PC1: principal component 1; PC2: principal component 2; CI−Di: integrated density class I; CII−Di: integrated density class II; CIII−Di: integrated density class III; CI−Dr: relative density class I; CII−Dr: relative density class II; CIII−Dr: relative density classes III; G: germination rate (G); RPSI: root protrusion speed index; SL: seedling length; UI: uniformity index; CVI: corrected vigor index; SDM: seedling dry matter; SA: severe alteration; MA: mild alteration; AA: absence of alteration.

Figure 6 shows the relationship between seed-tissue density and seedling performance. High-density tissue seeds generated more opaque radiographs (higher radiopacity), represented by regions of green, yellow, and red color that indicate high relative density. Thus, CI seeds showed a greater area occupied by colors closer to red, developing more vigorous seedlings (Figure 6A). On the other hand, low-density seeds (CIII) generated radiographs with darker regions (white, blue, and purple), which covered large seed areas, giving rise to abnormal seedlings (Figure 6D) or dead seeds (Figure 6E). Seeds that were deteriorated and/or physically damaged had lower tissue densities, corresponding to higher germination when the embryonic axis was not compromised, but poorer seedling development (Figure 6B,C).

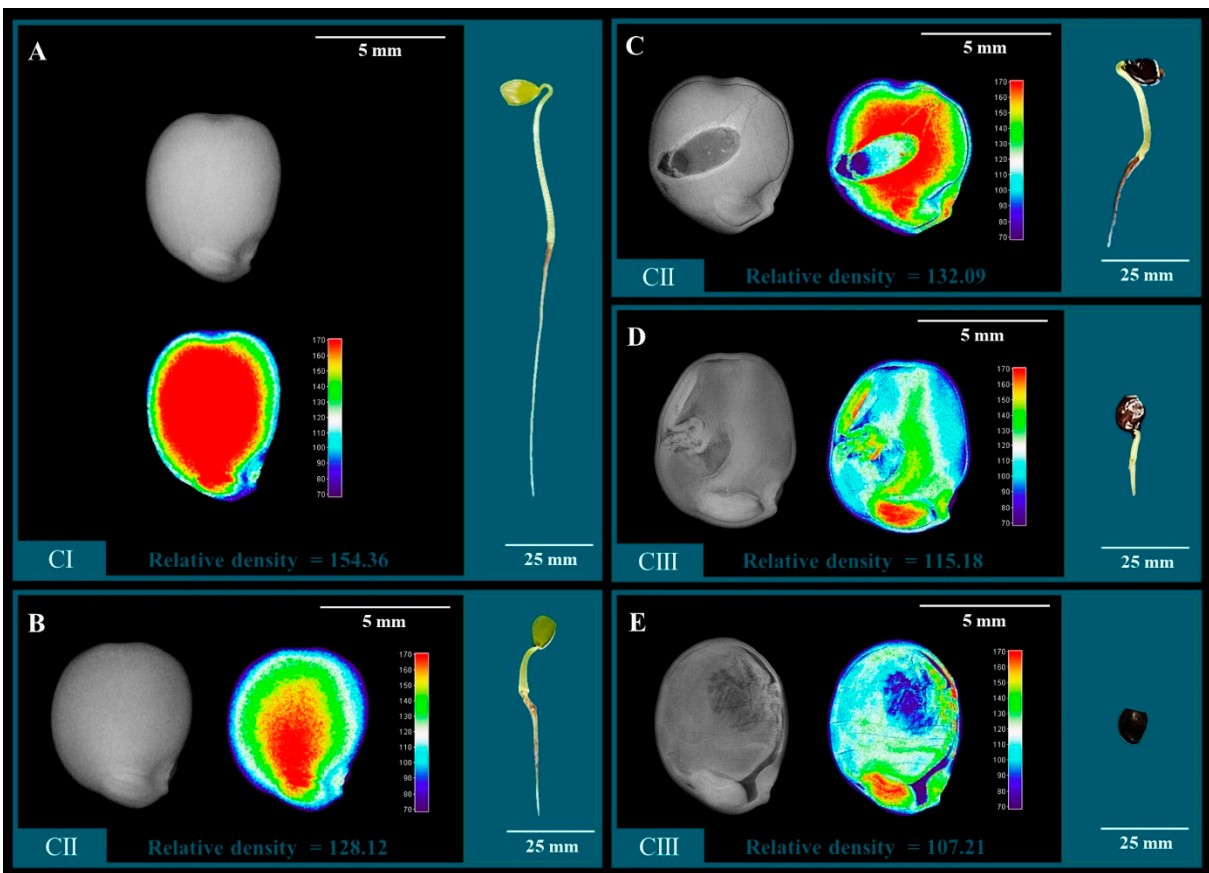

**Figure 6.** (**A–E**) Radiographs of *Senna macranthera* seeds and respective color representations of density and seedlings (7 days after sowing). CI: class I (relative density above 143 gray/pixel); CII: class II (densities from 118.1 to 143 gray/pixel); CIII: class III (densities below or equal to 118 gray/pixel).

### 3.2. Reforested Biomass Estimation and Carbon Stock

The results related to *Senna macranthera* diameter at ground level (DGL), height, carbon stock, and biomass are shown in Figure 7. Regarding the analyzed growth parameters, a greater diameter and height were verified in trees aged 66 months compared to trees aged 42 months. Similar results were also observed for carbon stock and biomass, with average increments of 26.96 kg (kg) and 57.35 kg, respectively, in trees aged 66 months. There were no expressive variations in the average values of carbon stocks and biomass in trees at 90 and 102 months of age. However, it is worth mentioning that the ages of the trees in this plantation characterize an area that is still early in the restoration process. Considering all sampling units, the carbon-stock variation ranged from 9.45 kg to 35.36 kg per tree, with an average value of 21.42 kg per tree (Figure 7).

### 3.3. Survival Rate and Replanting Cost Estimate

The costs of seedlings associated with planting and replanting *Senna macranthera* are listed in Table 3. Initial planting costs are 2500 seedlings × USD 3.10/seedling (BRL 15.965/seedling) which equals USD 7750/ha (BRL 39,912.50/ha). The highest cost of re-planting was associated with the highest mortality rate verified for planting at 42 months of age where there was a mortality rate of 81.25% at a combined cost of USD 6296.10/ha (BRL 32,424.92/ha). As seedling age increases from 42 to 102 months, the survival percentage was 45% or more. At 102 months and 75% seedling survival, the cost of replanting drops to USD 1937.50/ha (BRL 9978.13/ha). This translates to an average annual replanting cost of USD 6296.10 + 4262.50 + 1937.50 = 12,496.10/8 years = USD 1562.01 per year.

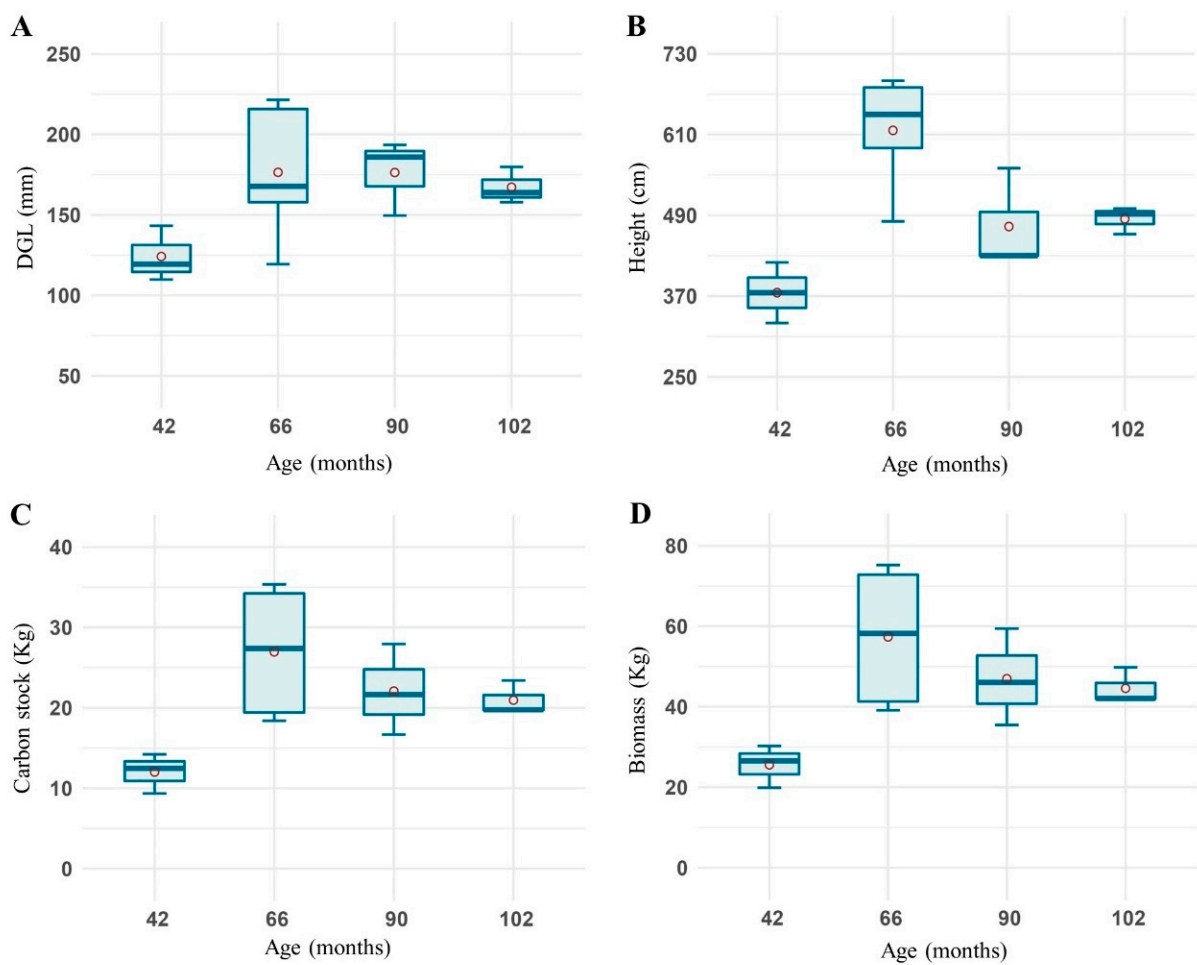

**Figure 7.** (**A**) Diameter at ground level (DGL), (**B**) height, (**C**) carbon stock, and (**D**) biomass in *Senna macranthera* trees of different ages (42, 66, 90, and 102 months).

**Table 3.** Survival and cost of planting and replanting seedlings of *Senna macranthera* (USD 1 = 5.15 reais (BRL)).

| Age (Years) | Age (Months) | Plant Survival (%) | Dead Plants (%) | Plants per Hectare (ha) | Cost of Seedlings | | Cost of Planted and Replanted Seedlings | |
|---|---|---|---|---|---|---|---|---|
| | | | | | USD/seedling | BRL/seedling | (USD/ha) | (BRL/ha) |
| 0.25 | 3 | n/a | n/a | 2500 | 3.10 | 15.965 | 7750.00 | 39,912.50 |
| 3.5 | 42 | 18.75 | 81.25 | 2031 | 3.10 | 15.965 | 6296.10 | 32,424.92 |
| 5.5 | 66 | 45 | 55 | 1375 | 3.10 | 15.965 | 4262.50 | 21,951.88 |
| 7.5 | 90 | 100 | 0 | 0 | 3.10 | 15.965 | - | - |
| 8.5 | 102 | 75 | 25 | 625 | 3.10 | 15.965 | 1937.50 | 9978.13 |

### 3.4. Cost Estimate for Selecting Higher Quality Seed

In order to process 2,088,000 seeds per year with a team of two full-time workers using an X-ray, a camera, and a computer in rented lab space, the total annual costs and the estimated annual revenue needed to cover both variable and fixed costs is USD 22,252 or USD 10,657.30 per full-time worker equivalent. Assuming that 81% of processed *Senna macranthera* seed is viable, the total cost per seed is USD 0.0132 (Table 4). The costs per seed range from USD 0.0112 to USD 0.0213 assuming that viable seeds are 95% and 50% of the total processed seeds, respectively. We estimated the cost of young seedlings as 15.38% (2 weeks/13 weeks) of the cost of a mature seedling planted at 3 months (13 weeks).

Assuming mature seedlings are USD 3.10/seedling, young seedlings were estimated at USD 0.477/seedling, compared to USD 0.0132/seed at an 81% germination rate.

**Table 4.** Production costs (USD) of radiographic imaging of *Senna macranthera* seeds assuming 81% of processed seeds are viable.

| | Total | Per Seed | Per Worker |
|---|---|---|---|
| Seeds Processed/Year | 2,088,000 | 2,088,000 | 1,000,020 |
| Viable Seeds Produced/Year (81%) | 1,691,280 | 1,691,280 | 810,016 |
| Full-Time Worker Equivalents | 2.09 | - | - |
| Price (USD/seed) | 0.013 | - | - |
| Annual Revenue | 22,252 | 0.0132 | 10,657.30 |
| Annual Operating Expenses | | | |
| Labor at USD 4.50/hour | | | |
| X-ray & Camera for Image | 9396 | 0.0056 | 4500.09 |
| Analyze & Separate Viable Seed | 9396 | 0.0056 | 4500.09 |
| Total Operating Expenses | 18,792 | 0.0111 | 9000.18 |
| Annual Ownership Expenses | | | |
| Depreciation and Interest | | | |
| Processing Equipment | | | |
| X-Ray (USD 25,000 at 15-year useful life) | 1667 | 0.0010 | 798.23 |
| Camera (USD 700 at 15-year useful life) | 47 | 0.00003 | 22.35 |
| Computer (USD 1000 at 15-year useful life) | 67 | 0.00004 | 31.93 |
| Lab Rent (12 months at USD 700/month) | 1680 | 0.0010 | 804.61 |
| Total Ownership Expenses | 3460 | 0.0020 | 1657.12 |
| Total Annual Cost | 22,252 | 0.0132 | 10,657.30 |
| Net Firm Income (NFI) | 0 | 0 | 0 |
| Return over Variable Cost (ROVC) | 3460 | 0.0020 | 1657.12 |
| Performance Measures | | | |
| Breakeven Revenue | | USD/seed | USD/worker |
| Long-run to Cover All Costs | | 0.0132 | 10,657.30 |
| Short-run to Cover Operating Costs | | 0.0111 | 9000.18 |

The seed cost in order to plant one hectare of *S. macranthera* seedlings was 2500 seed × USD 0.0132 = USD 33. This was only 0.43% of the estimated cost per hectare (USD 7750) of planting during the establishment year (Table 3). For replanting trees that are lost up to 8 years after establishment, annual costs of just seed alone per hectare (if incurred) ranged from USD 8.25 to 26.81, which was only 0.11% to 0.35% of replanting costs per hectare (USD 1937.50 to 6296.10) per year (Table 3).

## 4. Discussion

### 4.1. Implications for Seed Viability

The moisture content of seeds influences their optical density [30] and, hence, radiographic image quality. This may compromise the accuracy of results and visualization of seed internal morphology. In *Leucaena leucocephala* [31] and *Senna siamea* [15] seeds, tissue-density-related parameters could be compared through radiographs, and their moisture levels ranged from 8.1% to 8.8% and from 11.5% to 12.9%, respectively. Moisture uniformity among the analyzed lots is another aspect that must be considered for standardized evaluations and safe comparisons of seed physiological quality [32].

Semiautomated analysis of radiographs allowed us to identify differences in relative and integrated density among seed lots rather than X-ray absorption by *Senna macranthera* seed tissue. Higher tissue densities indicate greater resistance to X-ray passage, providing higher radiopacity (light) in digital radiographs. Integrated density is calculated considering the gray values of each image pixel, differing from relative density calculations that consider seed area [13]. Recently, seed research has reported the potential of both parameters to evaluate seed lots for different species [11,14–16]. However, although the results of physical parameters generated by X-ray analysis are effective in differentiating seed lots,

they must be compared to physiological tests in order to accurately infer their potential relationship with seed physiological quality [13].

As observed in this study, tissue density assessed by radiography was directly related to the physiological quality tests of *Senna macranthera* seeds. In general, these findings were more evident for lots 1, 2, and 9, which showed a higher proportion of dense seeds (CI) combined with higher physiological quality (germination and vigor). On the other hand, lot 8 showed a higher proportion of seeds with lower density (CIII) combined with lower physiological quality.

The relationship between physical attributes from X-ray testing and physiological quality has been reported in seeds of several forest species. For example, prior research [16] demonstrated that the X-ray technique was efficient in assessing the physical integrity of seed tissues and relating this to the physiological quality of *Piptadenia gonoacantha* seeds. This research was similar to this study, where seeds with lower tissue density had a higher proportion of internal physical damage and lower physiological quality. In our study, this was efficiently demonstrated by PCA, with a high correlation between severe alteration and class CIII (lower density) variables, as well as the proximity of their vectors with lot 8, which showed a physiological quality inferior to other lots. According to prior research [33], low tissue densities are related to severe morphological damage, deteriorated tissues, and unviable seeds. Positive correlations between tissue density and physiological quality of *Senna siamea* seeds were observed by Silva et al. (2020) [15], suggesting X-ray testing was efficient for the selection of seed lots for this particular species.

Tissue-density reduction and low physiological quality of seeds have been attributed to several causes, such as embryonic malformation in *Moquiniastrum polymorphum* [34], cracks in *Hancornia speciosa* [35], and fungal contamination and insect predation in *Leucaena leucocephala* [11], among others. Moreover, lower tissue density may be related to a reduced amount of energy reserves in tissues [36,37], which directly contributes to reduced seed germination and vigor. In this context, dividing seeds into density classes can be used in quality-control programs, allowing for the disposal of low-tissue-density seeds and improving the physical and physiological quality of seed lots.

It is worth noting that the X-ray technique helps to visualize, in detail, the internal morphology of seeds, which can help identify malformations, anomalies, mechanical damage, or insect injuries, in addition to tissues in an advanced state of deterioration [8]. However, to obtain the relationship between viability and vigor, a comparison with seedling development is required [10]. Therefore, in this study, we verified that seed lots with higher proportions of low-density classes produced smaller seedling lengths.

Most seeds with damage to the embryonic axis or with more than 50% area that was compromised produced dead seeds. Some seeds with mild alterations such as minor damage to cotyledons, which at first would not compromise viability and mobilization of seed reserves, generated abnormal seedlings or unviable seeds. This can be explained by the presence of pathogens such as some fungi, which are harmful to germination and seedling development but may not be detected via X-ray. Moreover, some physiological changes, such as intermediate phases of the deterioration process, are not identified by simply viewing radiographs [8,38]. Such alterations may cause changes in the reserve metabolism [32], and, hence, X-ray absorption by seed tissues, highlighting the importance of methods to measure tissue density in a precise and standardized way.

Finally, another detail observed in this study was the variability in tissue density, germination rate, and seed vigor among the different *Senna macranthera* matrices, despite being from the same region. The *Senna macranthera* matrices present genetic diversity. Prior research [39] validated an X-ray method for internal and densitometric analysis of *Lecythis pisonis* seeds and reported high genetic divergence among matrices. According to these researchers, such findings are important and can be applied to genetic improvement programs, where wide diversity enables the selection of superior genotypes by segregating successive generations.

*4.2. Carbon Offset Implications Planting in Degraded Areas and Survival Rate*

The results of the carbon stock and biomass analysis presented in this research corroborate the findings of Morais Junior et al. (2019) [26], showing the potential of *Senna macranthera* for use in carbon offset projects in degraded areas of the Brazilian Atlantic Forest. Obtaining information on carbon stock and biomass in native tree species that have not yet been studied is important in Brazil which leads to projections for the generation of carbon credits aimed at meeting the goals of the 2015 Paris Agreement on climate change [40,41]. In addition, it is important to conduct studies that evaluate the potential of native forest species for reforestation, conservation of natural resources, and mitigation of climate change.

The Brazilian Atlantic Forest is one of the most diverse forests in the world, but it is the most devastated biome in Brazil with only 27.3% of the area covered by native vegetation [42]. Therefore, sustainable management practices and plantations with species native to this biome are necessary for forest recovery in the long term [43]. According to Janishevski et al. (2015) [44], restoration programs should prioritize the use of native forest species in order to preserve biodiversity. *Senna macranthera* is a native Brazilian pioneer species found in the Atlantic Forest, with high initial growth rates, which enables faster regeneration [26]. In addition, it has low demand for high soil fertility [45].

When it comes to the survival rate of *Senna macranthera*, it was possible to verify considerable variations over the years, estimating variable costs of replanting during the period studied. Differences in survival rates may be related to several factors such as the distribution and diversity of chosen species, attack by leaf-cutting ants, and water shortages for different periods among other reasons [26]. In addition, the quality of the seeds and, consequently, the seedlings affect the survival of the species. Restoring degraded lands requires huge volumes of tree-planting material. Seed systems encompassing all forest reproductive material (e.g., seeds, cuttings, stakes, and wildings) are key to ensuring sufficient planting material with a diverse range of suitable species [22]. In this way, the use of vigorous seeds is crucial for the future success of plantings recovering and restoring native habitat in the Brazilian Atlantic Forest. Seeds are a critical and limited resource for restoring biodiversity and ecological function in degraded and fragmented ecosystems [46].

To meet the Brazilian demand for the restoration of 12.5 million hectares by 2030, it has been estimated that there will be a need for approximately 881 to 4443 metric tons of seeds per year, depending on the restoration technique [47]. This research also reports that these values are still far from the productive reality of the forest-seed sector, which is mostly small-scale, irregular, and lacking financial incentives. Applicable laws requiring minimum guidelines and quality control are the foundation for an ideal seed and seedling supply system [48]. However, the elaboration of uniform rules for evaluating the quality of native forest seeds has been slow due to the fact that they are economically less important species, as well as the lack of research that helps to define these rules [49].

Therefore, it is essential to have close integration between professionals and researchers who work in the areas of seed technology and forestry. Strong policies are needed to regulate the production, certification, and quality control of forest seeds in order to create opportunities for the trade of seeds and seedlings with better quality standards. Our findings demonstrate the importance of the X-ray technique and variables related to tissue density for quick, efficient, and nondestructive selection of *Senna macranthera* seeds. Therefore, this technique can be used to evaluate the physiological quality of seeds in order to produce seedlings for reforestation programs involving this species.

## 5. Conclusions

The X-ray technique was successfully used to select higher-quality seeds. Semiautomated analysis of radiographic images identified tissue-density variables related to the physiological quality of *Senna macranthera* seeds. The technique used in this study can help in selecting the most suitable seeds for the production of seedlings with a better quality standard and greater potential for survival after planting in the field. Since there is no

formal market for *Senna macranthera* seed, our estimated cost of USD 0.0132/seed at an 81% rate of germination is a baseline for future nonmarket and market valuations of this seed for reforestation. The annual cost of planting *S. macranthera* seedlings was USD 7750 per hectare during the establishment year. Replanting lost transplants during the third, fifth, seventh, and eighth years after initial planting averaged USD 3124 per year, ranging from USD 0 to 6296 per year.

*Senna macranthera* can enhance carbon storage in carbon-offset programs, providing a viable reforestation option for recovery projects in degraded areas in Brazil. Brazil's national targets for forest restoration have generated demand for the production of native seeds in terms of both quantity and quality. It is important to use seeds with high genetic and physiological quality for establishment success during reforestation using nondestructive, semiautomated, efficient, and fast seed-selection techniques. Thus, future scientific research should consider the possibility of expanding the technique used in this study to other native species in order to better restore native habitats in Brazil and elsewhere.

**Author Contributions:** Data collection, writing, methods, formal analysis, figures—J.d.O.A., D.T.P., G.B.Q., J.M.S., V.T.M.d.M.J., S.J.S.S.d.R., and A.K.H.; review, editing, supervision, and financial support—A.K.H. and D.C.F.d.S.D. All authors have read and agreed to the published version of the manuscript.

**Funding:** This study was financed in part by the Coordenação de Aperfeiçoamento de Pessoal de Nível Superior—Brasil (CAPES)—Finance Code 001, Conselho Nacional de Desenvolvimento Científico e Tecnológico—Brazil (CNPq), and Fundação de Amparo à Pesquisa do Estado de Minas Gerais—Brazil (FAPEMIG).

**Institutional Review Board Statement:** Not applicable.

**Informed Consent Statement:** Not applicable.

**Data Availability Statement:** The data summarized in this study are available on request from the corresponding author. All radiographic images are not publicly available due to the volume of images taken.

**Acknowledgments:** We are grateful to the Laboratório de Sementes (UFV) structure for seed analysis and the Departamento de Entomologia (UFV) for providing the use of X-ray equipment. We thank the three anonymous reviewers whose comments and edits improved the quality of this work.

**Conflicts of Interest:** The authors declare no conflict of interest. Supporting entities had no role in the design of the study; in the collection, analyses, or interpretation of data; in the writing of the manuscript, or in the decision to publish the results.

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
