# Peer review of "Selection of Superior Senna macranthera Seeds, Carbon Stock, and Seedling Survival, and Costs for Habitat Restoration"

_sustainability, doi:10.3390/su15139875_

Round 1
Reviewer 1 Report
Comments on the manuscript:
Saving Superior Seeds: Using Radiographic Images to Select High Physiological Seed Quality of Senna macranthera (DC. Ex Collad.) H.S. Irwin & Barneby for Woody Plant Restoration
Title is too long and does not reflect the whole content of the manuscript. The title only mentions the technique to evaluate the physiological quality of the seed and does not mention anything about the estimation of biomass, carbon stock, survival rate, estimation of seedling survival rate, replanting costs, and seed production.
Like the title, abstract does not have all the information presented in the manuscript.
Neither in the introduction is there any mention of the above-mentioned issues.
Hypotesis is missed
There is no parallelism between the title and the objective.
Material and Methods
The criteria used for the selection of the 10 mother trees are not explained.
They indicate that the seeds were hand scarified but do not indicate how or what they used for scarification.
They indicate that seedlings of length were imaged in a closed MDF box...What is an MDF box?
The authors use methodology as a synonym for techniques or protocols.
Etymologically methodology is the science that studies methods in general.
Results
It is incorrect to use the dot and the -1 in the units. one or the other should be used, but not both, it is redundant.
The estimation of biomass, carbon stock, survival rate, estimation of seedling survival rate, replanting costs, and seed production should be removed from the manuscript
Discussion
The estimation of biomass, carbon stock, survival rate, estimation of seedling survival rate, replanting costs, and seed production should be removed from the manuscript
Conclusions are very extensive and the objective of the work is repeated in the final part. They should be more punctual according to the objective, since it seems that a lot of detail is given.
The writing needs to be reviewed by an English expert
Author Response
Comments on the manuscript:
Saving Superior Seeds: Using Radiographic Images to Select High Physiological Seed Quality of Senna macranthera (DC. Ex Collad.) H.S. Irwin & Barneby for Woody Plant Restoration
Title is too long and does not reflect the whole content of the manuscript. The title only mentions the technique to evaluate the physiological quality of the seed and does not mention anything about the estimation of biomass, carbon stock, survival rate, estimation of seedling survival rate, replanting costs, and seed production.
We have shortened the title and made it more relevant to the whole content of the manuscript.
Like the title, abstract does not have all the information presented in the manuscript.
We have added more information to the manuscript abstract.
Neither in the introduction is there any mention of the above-mentioned issues.
We have improved the information in the manuscript introduction.
Hypothesis is missed
We have added a sentence in the last paragraph of the Introduction section on hypotheses.
There is no parallelism between the title and the objective.
By changing the title, the title is now more in line with our research objectives.
Material and Methods
The criteria used for the selection of the 10 mother trees are not explained.
The selection criteria were healthy trees aged 8.5 years. We added this information in the text.
They indicate that the seeds were hand scarified but do not indicate how or what they used for scarification.
Thanks for the observation. The seeds were mechanically scarified with sandpaper (number 100) on the opposite side of the embryonic axis. We added this information in the text.
They indicate that seedlings of length were imaged in a closed MDF box...What is an MDF box?
MDF is the abbreviation for medium density fiberboard. We added this information in the text.
The authors use methodology as a synonym for techniques or protocols. Etymologically, methodology is the science that studies methods in general.
We have replaced the word “methodology” with “methods” in three places in the writing to make this more clear.
Results
It is incorrect to use the dot and the -1 in the units. one or the other should be used, but not both, it is redundant.
Thank you for that. We have corrected the manuscript units.
The estimation of biomass, carbon stock, survival rate, estimation of seedling survival rate, replanting costs, and seed production should be removed from the manuscript
We respectfully disagree with your observation. Considering the journal's focus and scope (Sustainability), we believe the presence of these variables in the manuscript is essential. Furthermore, we found a good relationship between these data and seed data (physical and physiological). In our point of view, this contributed considerably to the relevance of the article in an unprecedented and multidisciplinary approach. Therefore, we chose to keep these analyses and data.
Discussion
The estimation of biomass, carbon stock, survival rate, estimation of seedling survival rate, replanting costs, and seed production should be removed from the manuscript
Please see previous response to comment directly above.
Conclusions are very extensive and the objective of the work is repeated in the final part. They should be more punctual according to the objective, since it seems that a lot of detail is given.
We have streamlined the Conclusions as requested.
The writing needs to be reviewed by an English expert
One of the co-authors is a native English speaker and has reviewed the manuscript again for English.
Submission Date
27 February 2023
Date of this review
27 Mar 2023 21:21:02
Reviewer 2 Report
Dear authors,
The title „Saving Superior Seeds: Using Radiographic Images to Select 2 High Physiological Seed Quality of Senna macranthera (DC. Ex 3 Collad.) H.S. Irwin & Barneby for Woody Plant Restoration” is sugestive and siutable for the content of this work.
First, I would like to thank you for presenting your results. Overall, I found that the topic of your paper is very interesting and current. The study design setup and the applied methods are adequate and the analyses were well performed. The article is understandably written and well-organized, contain all the components I would expect, and the sections are well-developed. The methods are quite well explained, the results are well described, and the discussion is carried out very well. So, I think you did a good job.
Best regards!
Author Response
Comments and Suggestions for Authors
Dear authors,
The title „Saving Superior Seeds: Using Radiographic Images to Select 2 High Physiological Seed Quality of Senna macranthera (DC. Ex 3 Collad.) H.S. Irwin & Barneby for Woody Plant Restoration” is sugestive and siutable for the content of this work.
First, I would like to thank you for presenting your results. Overall, I found that the topic of your paper is very interesting and current. The study design setup and the applied methods are adequate and the analyses were well performed. The article is understandably written and well-organized, contain all the components I would expect, and the sections are well-developed. The methods are quite well explained, the results are well described, and the discussion is carried out very well. So, I think you did a good job.
Thank-you very much for taking the time to review the manuscript and for your review!
Submission Date
27 February 2023
Date of this review
04 Apr 2023 15:52:31
Reviewer 3 Report
I completed the evaluation of title manuscript " Saving Superior Seeds: Using Radiographic Images to Select High Physiological Seed Quality of Senna macranthera (DC. Ex Collad.) H.S. Irwin & Barneby for Woody Plant Restoration" I reviewed the manuscript based on originality, technical quality, clarity of presentation and potential significance. I believe that the manuscript has valuable data, however the manuscript has some mistakes terminology, and its goal, methodology and discussions have fluently linked between introduction and results, it is easy to fallow. The conclusion is written too long. If the authors revise the conclusion to include more summary and important information, it will make the manuscript easier to understand.
Required corrections and suggestions were shown with yellow color on the manuscript.
Best Regards

Author Response
Comments and Suggestions for Authors
I completed the evaluation of title manuscript " Saving Superior Seeds: Using Radiographic Images to Select High Physiological Seed Quality of Senna macranthera (DC. Ex Collad.) H.S. Irwin & Barneby for Woody Plant Restoration" I reviewed the manuscript based on originality, technical quality, clarity of presentation and potential significance. I believe that the manuscript has valuable data, however the manuscript has some mistakes terminology, and its goal, methodology and discussions have fluently linked between introduction and results, it is easy to fallow. The conclusion is written too long. If the authors revise the conclusion to include more summary and important information, it will make the manuscript easier to understand.
Required corrections and suggestions were shown with yellow color on the manuscript.
Best Regards
Submission Date
27 February 2023
Date of this review
14 Apr 2023 11:49:12
Thanks for the review and suggestions. They contributed to the improvement of the manuscript! We accepted most of your suggestions and made corrections to the manuscript. We have also responded to comments in the pdf file that you sent and have included this revised pdf file uploaded to the MDPI Sustainability system.

Round 2
Reviewer 1 Report
Comments on the manuscript:
Saving Superior Seeds: Using Radiographic Images to Select High Physiological Seed Quality of Senna macranthera (DC. Ex Collad.) H.S. Irwin & Barneby for Woody Plant Restoration
Materials and methods. In materials they do not indicate which are the treatments
Results: In Figure 3, the title of the ordinate says proportion. Proportion is different than percentage. It is need to change it.
In figure 7, it is necessary to say that it is DAS
The writing needs to be reviewed by an English expert
Author Response
Comments and Suggestions for Authors
Comments on the manuscript: Saving Superior Seeds: Using Radiographic Images to Select High Physiological Seed Quality of Senna macranthera (DC. Ex Collad.) H.S. Irwin & Barneby for Woody Plant Restoration
Materials and methods. In materials they do not indicate which are the treatments
We have added this information.
Results: In Figure 3, the title of the ordinate says proportion. Proportion is different than percentage. It is need to change it.
Thank you for that. We corrected the title of the ordinate.
In figure 7, it is necessary to say that it is DAS
DAS is diameter at ground level. We changed the acronym to DGL and specified its meaning. Diameter at ground level (DGL).
The writing needs to be reviewed by an English expert
Dr. Aaron Kinyu Hoshide (one of the co-authors on the manuscript) is a U.S.A. citizen and a native English speaker. He has edited the entire manuscript for English.
Submission Date
27 February 2023
Date of this review
25 May 2023 00:02:34
Reviewer 3 Report
Dear Editor,
The authors made the necessary corrections. It is appropriate to publish.
Best Regards....
Author Response
Comments and Suggestions for Authors
Dear Editor,
The authors made the necessary corrections. It is appropriate to publish.
Best Regards....
Thanks very much for your comments and edits as they have significantly improved the quality of our work!
Submission Date
27 February 2023
Date of this review